



# The diurnal and seasonal variability of ice nucleating particles at the High Altitude Station Jungfraujoch (3580 m a.s.l.), Switzerland

Cyril Brunner[1], Benjamin T. Brem[2], Martine Collaud Coen[3], Franz Conen[4], Martin Steinbacher[5], Martin Gysel-Beer[2], and Zamin A. Kanji[1]

[1]Institute for Atmospheric and Climate Science, ETH, Zurich, CH-8092, Switzerland
[2]Laboratory of Atmospheric Chemistry, Paul Scherrer Institute, CH-5232 Villigen PSI, Switzerland
[3]Federal Office of Meteorology and Climatology, MeteoSwiss, CH-1530 Payerne, Switzerland
[4]Department of Environmental Sciences, University of Basel, CH-4056 Basel, Switzerland
[5]Laboratory for Air Pollution / Environmental Technology, Empa, Überlandstrasse 129, CH-8600 Dübendorf, Switzerland

**Correspondence:** C. Brunner (cyril.brunner@env.ethz.ch) and Z.A. Kanji (zamin.kanji@env.ethz.ch)

**Abstract.** Cloud radiative properties, cloud lifetime, and precipitation initiation are strongly influenced by the cloud phase. Between $\sim$ 235 and 273 K, ice nucleating particles (INPs) are responsible for the initial phase transition from the liquid to the ice phase in cloud hydrometeors. This study analyzes immersion-mode INP concentrations measured at 243 K at the High Altitude Research Station Jungfraujoch (3580 m a.s.l.) between February 2020 and January 2021, thereby presenting the longest continuous, high-resolution (20 min) data set of online INP measurements to date. The high time resolution and continuity allow to study the seasonal and the diurnal variability of INPs. After exclusion of special events, like Saharan dust events (SDEs), we found a seasonal cycle of INPs, highest in April (median in spring 3.1 INP std $L^{-1}$), followed by summer (median: 1.6 INP std $L^{-1}$) and lowest in fall and winter (median: 0.5 INP std $L^{-1}$ and 0.7 INP std $L^{-1}$, respectively). Pollen or subpollen particles were deemed unlikely to be responsible for elevated INP concentrations in spring and summer, as periods with high pollen loads from nearby measurement stations do not coincide with the periods of high INP concentrations. Furthermore, for days when the site was purely in the free troposphere (FT), no diurnal cycle in INP concentrations was observed, while days with boundary layer intrusions (BLI) showed a diurnal cycle. The seasonal and diurnal variability of INPs during periods excluding SDEs is with a factor of 7 and 3.3, respectively, significantly lower than the overall variability observed in INP concentration including SDEs of more than three orders of magnitude, when peak values result from SDEs. The median INP concentration over the analyzed 12 months was 1.2 INP std $L^{-1}$ for FT periods excluding SDEs, and 1.4 INP std $L^{-1}$ for both FT and BLI, and incl. SDEs, reflecting that despite SDEs showing strong but comparatively brief INP signals, they have a minor impact on the observed annual median INP concentration.

## 1 Introduction

The ratio of ice crystals to liquid droplets in a cloud strongly determines its radiative properties, lifetime, and precipitation initiation (e.g., Lau and Wu, 2003; Lohmann and Feichter, 2005; Hoose and Möhler, 2012; Murray et al., 2012; Mülmenstädt et al., 2015; Heymsfield et al., 2020). For cloud hydrometers with a temperature between $\sim$ 235 and 273 K, the phase transition



from the metastable liquid phase to the ice phase is initially supported by ice nucleating particles (INPs, Pruppacher and Klett, 1997; Vali et al., 2015). However, ice nucleation on INPs remains insufficiently understood and quantified (Murray et al., 2021). Knowledge about a diurnal and seasonal variability can help to better understand the sources and sinks of atmospheric INPs, but

has only been addressed in a handful of studies. Conen et al. (2015) analyzed the seasonal INP fluctuation at the High Altitude Research Station Jungfraujoch (JFJ) over a year (June 2012 - June 2013) with 24-hour filter samples and their subsequent analysis with immersion freezing assays. They evaluated INPs active at 265 K and observed highest INP concentrations during June, July and August ($10^{-3}$ to $10^{-4}$ $L^{-1}$) and lowest between January and the beginning of April with $10^{-5}$ to $10^{-4}$ $L^{-1}$, which correlated with ambient temperature. Wex et al. (2019) analyzed the seasonality in INP number concentrations at four

different locations in the Arctic using filter samples, where each filter was collecting ambient particles over four days to two weeks. They found a seasonal trend, with INP concentrations lowest in winter and highest in spring. At 260 to 266 K, the seasonal cycle in INP number concentrations spread up to three orders of magnitude ($\sim 10^{-5}$ to $10^{-2}$ $L^{-1}$). Schneider et al. (2020) studied the seasonal cycle of INPs in the boreal forest in Finland between March 2018 and May 2019, where the INPs were collected on a filter, which got replaced after 24 to 144 h, and analysed offline on a cold stage. At 252 K, they observed a

minimum INP concentration in wintertime, on the order of 1 INP std $L^{-1}$, and maxima in early and late summer of $\sim$4 INP std $L^{-1}$. The trend was more pronounced at warmer temperatures, but visible across the entire investigated temperature spectrum (265 - 252 K). Tobo et al. (2020) studied seasonal INP concentrations with 72-hour filter samples taken on a building 458 m above ground in Tokyo between August 2016 and July 2017, followed by testing the particle collected on the filters for their ice nucleation activity on a cold stage. They found a weak seasonal cycle for INPs active below 253 K, with spikes during

Asian dust events. For INPs active at warmer temperatures, specifically between 258 and 263 K, higher concentrations were recorded in the warm/wet season and lower concentrations in the cold/dry seasons. They attributed the seasonal trend to Asian dust events and seasonal variations in certain particles of biological origin linked to local meteorology. Schrod et al. (2020) analyzed the INP concentrations at four locations across the globe (the Amazon, Caribean, central Europe and the Arctic) for almost 2-years (May 2015 - January 2017). Electrostatic aerosol samplers were used to collect ambient particles onto silicon

wafers once every 2 days at noon for 50 min. The wafers were later exposed to a saturation ratio with respect to water of $0.95 \leq S_{\mathrm{w}} \leq 1.01$ at 253 K, 248 K, and 243 K and the number of formed ice crystals was used to deduce the mean INP concentration. They found average concentrations between sites to differ by less than a factor of 5. Short-term fluctuations dominated most of the total variability across all stations. To summarize, a seasonal dependence in INP concentrations was apparent in all these studies, where highest INP concentrations were observed in spring and summer. All these studies were performed using offline

sampling techniques, where changes to the ice nucleation ability of particles cannot be excluded between sampling and the analysis. Besides, they typically provide INP number concentrations at temperatures $\geq 250$ K, while online INP counters often struggle at temperatures $\geq 250$ K with the low abundance of atmospheric INPs because of their instruments' limit of detection (Cziczo et al., 2017). As the INP concentrations increase with decreasing temperature (e.g., see DeMott et al., 2010) online INP counter can measure statistically robust data from $\approx 250$ K down to 220 K and lower. Korolev et al. (2003) and Field

et al. (2004) showed using in-situ measurements, that, depending on the measurement location, approximately only half of the clouds at 253 and 258 K contain the ice phase, while warmer clouds are often completely free of the ice phase. This suggests



that in these clouds only INPs active at temperatures below 258 K are numerous enough to impact the cloud phase. Therefore, we aim to expand our knowledge about the seasonality in INPs to INPs active at a temperature of 243 K. In addition, our measurement location at the JFJ allows us to draw conclusions about INPs in both, free tropospheric as well as boundary layer

intruded air masses. In addition, offline techniques often require long sampling times resulting in too poor temporal resolution to capture diurnal variation within INP concentrations. Online measurement techniques provide the needed temporal resolution, yet required until 2019 an operator to be present at the site for the duration of the experiment to perform regular maintenance of the INP counters (Bi et al., 2019; Brunner and Kanji, 2021; Möhler et al., 2021). Therefore, there are only a few studies investigating the diurnal variability of INP concentrations (e.g., Isono et al., 1971; Rosinski et al., 1995), which, however, are

based on measurements of less than ten days. Only more recently, Wieder et al. (2021, in prep.) studied the diurnal cycle of INPs on a mountain top (2693 m a.s.l.) in Switzerland and simultaneously in a nearby valley over a period of two months in early spring. They found a diurnal cycle of INPs at the mountain top, lowest in the morning and highest at the beginning of the night. The cycle was attributed to orographic lifting from low elevation upstream the measurement site. There was no diurnal cycle at the valley measurement site apparent.

Here, we used the automated horizontal ice nucleation continuous chamber (HINC-Auto, Brunner and Kanji, 2021) to measure the INP concentration at a center lamina temperature of $T = 243.15$ K and $S_w = 1.04$ at the JFJ between February 7, 2020 and January 31, 2021. The long duration, together with a sampling interval of 20 minutes allowed us to study the seasonal, and the diurnal variability of INP concentrations at the sampling site.

## 2   Materials and methods

The Sphinx observatory is located on the Jungfraujoch (JFJ, 46.330° N, 7.590° E), a saddle between Mt. Mönch and Mt. Jungfrau in the Swiss alps. The JFJ has a long track record of experimental field studies on atmospheric aerosols and their interaction with clouds (Bukowiecki et al., 2016). With an altitude of 3580 m a.s.l., it is often located in the free troposphere (FT). Occasionally, air masses from the planetary boundary layer (PBL) are lifted to or mixed in the ambient air present at JFJ. These boundary layer intrusions (BLI) are most frequently observed in summer and during day time (Collaud Coen et al.,

2011). The method used to estimate the air mass type present at the site during measurements is discussed later in subsection 2.2. Furthermore, the remote location allows to study background concentrations of atmospheric aerosol, like INPs. However, in 2020 and 2021 frequent construction work at Jungfraujoch caused intermittent episodes of local pollution (Bukowiecki et al., 2021), which made a subsequent filtering of the data for polluted episodes necessary (see section 2.1). The JFJ site is part of the Global Atmospheric Watch (GAW) program, the pan-European Aerosol, Clouds and Trace Gases Research Infrastructure

(ACTRIS), the Swiss National Air Pollution Monitoring Network (NABEL), and the Swiss meteorological network (Swiss-MetNet). As such, a number of atmospheric measurements are continuously run at the JFJ, amongst others, the total particle number concentration (condensation particle counter (CPC), TSI 3772, lower cut-off size: 14 nm) and size distribution (scanning mobility particle sizer (SMPS); optical particle sizer (OPS), TSI 3300; fine dust monitoring device, Fidas® 200), aerosol light absorption properties (aethalometer, MAGEE scientific AE33), and aerosol light scattering and backscattering properties



(nephelometer, Airphoton IN101) as well as meteorological standard parameters (e.g., ambient temperature, relative humidity, atmospheric pressure, wind speed and direction). Meteorological standard parameters for the JFJ, precipitation rates, and pollen concentrations for other stations were obtained from the IDAWEB interface of MeteoSwiss (https://gate.meteoswiss.ch/idaweb, last accessed April 27, 2021).

## 2.1 INP measurements

INP concentrations are measured using an automated continuous flow diffusion chamber, the automated Horizontal Ice Nucleation Chamber (HINC-Auto, Brunner and Kanji, 2021). HINC-Auto sampled ambient air and measured the INP concentration at a center lamina temperature of $T$ = 243.15 K and at a supersaturated saturation ratio with respect to water of $S_\mathrm{w}$ = 1.04 between February 7, 2020 and January 31, 2021, in unites of INP std $\mathrm{L}^{-1}$ (per standard liter, normalized to $T$ = 273.15 K and an atmospheric pressure of $p$ = 1013.25 hPa). Ambient air was sampled via a heated ($T$ = 293.15 K) total aerosol inlet (Wein-
gartner et al., 1999), which feeds also the other aerosol measurements at the JFJ. Before entering HINC-Auto, the sampled air is dried using a diffusion dryer ($S_\mathrm{w} \leq$0.008 at 20 °C, filled with 4 Å-molecular sieve). The sampling volume rate was set to 0.283 std L $\mathrm{min}^{-1}$. The sampling interval was 20 min and consisted of sampling ambient air (15 min) and particle-free ambient air (5 min) between each measurement. In HINC-Auto, false-positive counts can arise due to frost breaking off the chamber walls. They are present during the measurement of ambient, as well as particle-free air. During the 5 min before and after
an ambient air measurement, the number of false-positive frost particles are counted and time-proportionally subtracted from the ambient air measurement to yield the background-corrected ambient INP concentrations. Because of Poisson statistics it is likely that during the time-normalized particle-free measurements more or fewer false-positive counts arise than during the time-normalized ambient air measurement. This leads to a variable systematic bias of the background-corrected ambient INP concentrations for each sampling interval even if the true atmospheric INP concentration was to remain constant. The standard
deviation of the resulting probability density function corresponds to the stated counting uncertainty of $\pm$ 1 $\sigma$, provided after each stated INP concentration in this work, and is on median $\pm$ 1.37 INP std $\mathrm{L}^{-1}$. The bias in background correction can lead to negative reported values of background-corrected ambient INP concentrations whenever the true INP concentration is close to or below the chamber background. These negative readings are retained in the data set. In addition, applying a moving average over 3 measurements greatly reduces the number of reported negative INP concentrations. For HINC-Auto, the count-
ing uncertainty is identical to the limit of detection (LOD). See Brunner and Kanji (2021) for a more detailed description of HINC-Auto and the derivation of the INP concentrations.

The measured INP concentrations are here statistically described by the median and $25^{th}$ to $75^{th}$ percentiles ($Q_{25\%}$ - $Q_{75\%}$). We assume atmospheric INP concentrations away from sources to be log-normally distributed, as proposed generally for atmospheric pollutants by Ott (1990) and supported by other studies (e.g., Welti et al., 2018; Schrod et al., 2020). Therefore,
an adequate statistical description would be the log-mean and log-standard-deviation. However, measurements reporting a negative INP concentration don't allow to include all data when calculating the logarithm. For log-normally distributed data without any skewness, the log-mean is identical to the median. Hence, we chose to report the median.





Frequent construction work at the JFJ during the observation period caused intermittent interference from local pollution (Bukowiecki et al., 2021). Therefore, the INP measurements had to be filtered (75.5% remain after filtering). High frequency
fluctuations in the total particle concentrations were observed during periods with pollution from the construction site. These fluctuations in the CPC and the $\geq 0.3$ µm - optical particle counter channel of HINC-Auto were used to obtain unpolluted INP measurements (N = 15843). A more detailed description of the data filtering process can be found in Brunner et al. (2021, in review).

## 2.2    Classification of air masses

### 2.2.1    Free tropospheric air masses and boundary layer intrusions

The distinction between undisturbed FT air masses and FT affected by boundary layer intrusions, hereafter simply referred to as BLI conditions, is made according to Brunner et al. (2021, in review). It is based on a combined criterion considering both the 222-Radon concentration ($^{222}$Rn, Griffiths et al., 2014, first factor in equation (1)) and the total number concentration of particles with diameters larger than $d \geq 90$ nm ($N_{90}$, Herrmann et al., 2015, second factor in equation (1)).Subsequently, the
probability ($P_{\mathrm{FT}}$) of the sampled air to represent undisturbed FT air mass is obtained using:

$$P_{\mathrm{FT}} = \frac{\mathrm{PDF}_{\mathrm{FT}}(^{222}\mathrm{Rn\ conc.})}{\mathrm{PDF}_{\mathrm{FT}}(^{222}\mathrm{Rn\ conc.}) + \mathrm{PDF}_{\mathrm{BLI}}(^{222}\mathrm{Rn\ conc.})} \frac{1}{1 + e^{k(\mathrm{N}_{90} - \mathrm{N}_{90,\mathrm{TH}})}} \tag{1}$$

where $\mathrm{PDF}_{\mathrm{FT}}$ and $\mathrm{PDF}_{\mathrm{BLI}}$ are the probability density functions with FT or BLI log-normal fit parameters, respectively, $k$ is the slope factor to capture the seasonality as discussed later, here $k = 0.1$, and $N_{90,\mathrm{TH}}$ is the $N_{90}$ threshold midpoint, here $N_{90,\mathrm{TH}}$ = 120 cm$^{-3}$. $\mathrm{PDF}_{\mathrm{FT}}$ and $\mathrm{PDF}_{\mathrm{BLI}}$ are inferred from from long-term radon measurements at the JFJ between January 1, 2009
and December 31, 2020, where the frequency distribution of the logarithm of the $^{222}$Rn concentrations, has a bimodal shape with two discernible but overlapping modes. The two normal distributions were fitted to this frequency distribution, where the modes at lower and higher concentrations are assumed to represent undisturbed FT and BLI conditions, respectively. The resulting probability density functions for $^{222}$Rn concentrations of FT and BLI air masses are according to:

$$\mathrm{PDF}_{\mathrm{FT}}(^{222}\mathrm{Rn\ conc.}) = \frac{1}{\sigma_{\mathrm{FT}}\sqrt{2\pi}} e^{-\frac{1}{2}\left(\frac{\log_{10}\left(^{222}\mathrm{Rn\ conc.}\right) - \mu_{\mathrm{FT}}}{\sigma_{\mathrm{FT}}}\right)^2} \tag{2}$$

where the log-normal fit parameters are $\mu_{\mathrm{FT}}$ = -0.139 Bq/std m$^3$, $\sigma_{\mathrm{FT}}$ = 0.239 Bq/std m$^3$ for $\mathrm{PDF}_{\mathrm{FT}}$, and $\mu_{\mathrm{BLI}}$ = 0.403 Bq/std m$^3$, $\sigma_{\mathrm{BLI}}$ = 0.238 Bq/std m$^3$ for $\mathrm{PDF}_{\mathrm{BLI}}$. Herrmann et al. (2015) showed, depending on the seasons at the JFJ, for values of $N_{90} \geq 100$ - 150 cm$^{-3}$ to be only present during BLI conditions. $N_{90}$ values below this threshold do not exclude BLI conditions. Therefore, the second factor of equation (1) forces $P_{\mathrm{FT}}$ to low probabilities whenever $N_{90}$ exceeds the defined threshold, while below the threshold $P_{\mathrm{FT}}$ becomes identical to using the first factor of equation (1) ($^{222}$Rn term) only. A more
detailed description can be found in Brunner et al. (2021, in review). $^{222}$Rn is measured at the JFJ according to Griffiths et al. (2014) and $N_{90}$ retrieved from SMPS measurements. The temporal resolution of the $^{222}$Rn measurements is one every 30 min., and one every 6 min. for the $N_{90}$ measurements.





### 2.2.2 Saharan dust events and background conditions

The JFJ is frequently influenced by Saharan dust events (SDEs; Collaud Coen et al., 2004). Between 2001 and 2017, 10 - 50

SDEs were reported each year with an annual total duration of 200 - 700 hours (Bader et al., 2021). To detect SDE periods the method of Brunner et al. (2021, in review) is applied. It uses four tracers as indicators for SDEs: the single scattering albedo Ångström exponent, measured and retrieved from nephelometer and aethalometer measurements at the JFJ (Collaud Coen et al., 2004), the attenuated backscatter of the Lufft CHM15k-Nimbus ceilometer, operated by MeteoSwiss at Kleine Scheidegg (KSE, 46.547° N, 7.985° E, 2061 m a.s.l., Hervo et al., 2016), 4.4 km north of the JFJ and 1500 m lower in altitude, particle

surface residence times over the Saharan desert of air parcels arriving at the JFJ, modelled using the Lagrangian particle dispersion model FLEXPART (Version 9.1_EMPA, Stohl et al., 2005), and the satellite-retrieved dust mass concentrations from the Copernicus Atmosphere Monitoring Service (CAMS). If at least one tracer indicates a SDE, the period is labeled as SDE. This conservative approach aims at providing a clean subset of data set for non-SDE cases, whereas the subset with SDE label may include some non-SDE periods. SDEs can occur in FT or BLI conditions, denoted as $FT_{SDE}$ and $BLI_{SDE}$, respectively.

In the total period analyzed, 31 SDEs were recorded with a total duration of 55 days and 20 hours. Periods without a SDE signal are labeled as background periods (BG = total - SDE). BG periods are further divided into FT and BLI periods ($FT_{BG}$ and $BLI_{BG}$, correspondingly). The conditions for a positive SDE-signal and further information can be found in Brunner et al. (2021, in review).

## 3 Results

The observed seasonal and diurnal variability of INP concentrations are discussed in the following sections. First, the total investigated period is brought into context with previous seasons with regard to standard meteorological parameters and the type of air mass present at the site. Then, the seasonal signature of different air masses is analyzed. Finally, the observed diurnal variability is presented for different air masses.

### 3.1 The seasonal INP variability at the JFJ

During February 2020 - January 2021, the mean temperature at the JFJ was -5.9 °C, representing the warmest period on record since 1933 and 2.0 °C warmer than February 1933 - January 1971. On February 10, the storm "Sabine" led to wind speeds of up to 54 m/s, which corresponds to the $99.6^{th}$ percentile of all daily maximum wind measurements at the JFJ. Yet, this is still well below the record wind velocity of 74 m/s, measured on January 6, 1998. The site was 40% less in clouds (where a relative humidity (RH) greater than 96% is considered as in-cloud) compared to previous years (2020-2021: 16.1% vs. 1970-

2019: 26.6%). Note that the prevalence of in-cloud conditions based on RH is underestimating the true prevalence of in-cloud conditions, but relative changes are well captured (e.g., see Herrmann et al., 2015). There were two exceptionally dry period in the end of March and the beginning of April (March 23 and April 4: RH = 2.2% in both cases at ambient temperatures of -10.3 °C and -8.5 °C, respectively, resulting in dew points of -47.2 and -48.8 °C, respectively). Ozone concentrations implied





no intrusion of stratospheric air masses (March 23 and April 4: ozone = 50.8 ppb and 61.0 ppb, respectively; stratospheric

intrusions often go along with ozone concentrations >70 ppb (Stohl et al., 2000)). During the late March/early April period, air masses were largely advected from high northern latitudes. With 38%, the JFJ experienced 5% more FT periods in relative terms than in the 2008-2019 period (36%). Figure 1 shows a more detailed representation of the air masses present during the analyzed period compared to historic observations. The fractions of days purely in FT, of days with a mix of FT and BLI, and of days only in BLI air masses for February 2020 - Jan 2021 were overall representative for the fractions observed during

previous years, as they rarely exceed the climatological $Q_{25\%}$ - $Q_{75\%}$ range. May and August 2020 had more days purely in the FT at the expense of days only with BLI for May. February and November 2020 showed a significantly higher fraction of mixed air masses while there were fewer days purely with BLI.

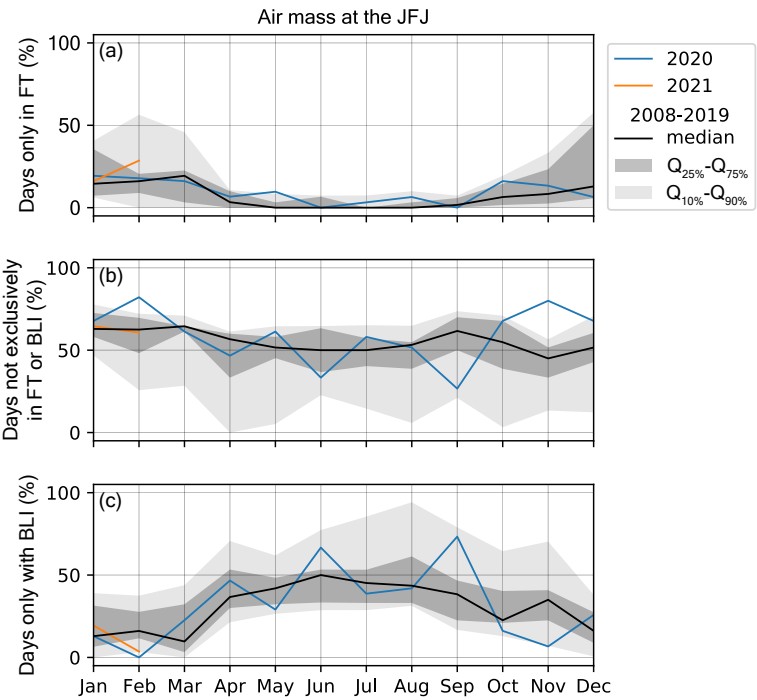

**Figure 1.** Fraction of days (**a**) entirely in FT air masses, (**b**) both in FT and BLI air masses, and (**c**) only in BLI air masses for 2020 (blue), Jan-Feb 2021 (orange) and 2008-2019 (gray). $Q_{10\%}$, $Q_{25\%}$, $Q_{75\%}$, and $Q_{90\%}$ correspond to the historic $10^{th}$, $25^{th}$, $75^{th}$, and $90^{th}$ percentiles between 2008 and 2019.

Figure 2a-c shows the monthly variation of the INP concentrations at the JFJ. The median INP concentrations are highest in April and July/August and lowest in December. Decomposing the data into periods with SDEs and BG conditions reveals for

the high $Q_{75\%}$-concentrations in March to be a result of SDEs, as four SDEs cumulatively made up 42% of the available time in March. Overall, the monthly median INP concentration during SDEs always exceeded the $Q_{75\%}$ of the BG concentrations. June had the most active SDE of the investigated period with a duration of 116 h and a $Q_{95\%}$ of 881.1 INP std L$^{-1}$. Together with two more brief and less active SDEs, June had the highest monthly INP concentrations during SDEs (median = 223.0 INP std





$L^{-1}$, $Q_{75\%}$ = 464.8 INP std $L^{-1}$), two orders of magnitude higher than the corresponding BG concentrations, consistent with
results from Chou et al. (2011). After June, February 2020 showed median highest INP concentrations during SDEs, followed
by March (median = 46.9 INP std $L^{-1}$ and 35.1 INP std $L^{-1}$, respectively). There were no recorded SDEs in December 2020
and January 2021. Overall Saharan dust contributed to 74.7 ± 0.2% of the total INPs observed at the sampling conditions at the
JFJ (see Brunner et al., 2021, in review). BG concentrations in June were lower compared to spring and July/August. Dividing
BG periods into $FT_{BG}$ and $BLI_{BG}$ shows similar distributions in both cases for most of the investigated period. Median and
$Q_{75\%}$ concentrations in April, July, and - to a lesser extent - in August were higher during $BLI_{BG}$ than in $FT_{BG}$ air masses.
Independent of the air mass, April showed the highest BG INP concentrations. In contrast to observations of INP active at
warmer temperatures (Conen et al., 2015; Schneider et al., 2020), no correlation between the ambient temperature and the
INP concentrations is evident in our data (Spearman's rank correlation coefficient = 0.149, $R^2$ = 0.012). We hypothesize that
because the investigated ice-activation temperature in our study (243 K) is distinctively colder than the ambient temperature
(median: 267 K) in contrast to Conen et al. (2015, INP conc. at 265 K, median ambient temperature = 266.5 K), in the latter
case a substantial fraction of the INPs can be removed from the air layers around the site when the INPs activate and the formed
ice crystals sediment. However, the observations by Schneider et al. (2020) do not support this hypothesis (INP conc. at 257 K,
median ambient temperature = 278 K). Comparing the observed seasonal pattern in total particle number concentrations (see
Fig. 2d-f) with the observed seasonality in INP concentrations, both concentrations are highest in spring and summer and lowest
in winter. However, while BG INP concentrations peaked in April for both, $FT_{BG}$ and $BLI_{BG}$, only a peak in the $75^{th}$ percentile
is apparent in April for the total particle concentration. While the drop in BG INP concentrations after the summer occurred
continuously in August and September, total particle number concentrations remained at summer levels also in September. The
seasonal median BG total particle concentrations varied by a factor of ~3, while the corresponding INP concentrations varied
by a factor of 22. On a more detailed level, April was the month with fifth highest total particle concentrations, with August and
May being the months with the highest loading. During the exceptionally dry period (March 23 - April 4), INP concentrations
were four times higher than before this period, but remained at the same level or increased further afterwards, suggesting for
the high INP concentrations in April are not connected to the mentioned dry period.



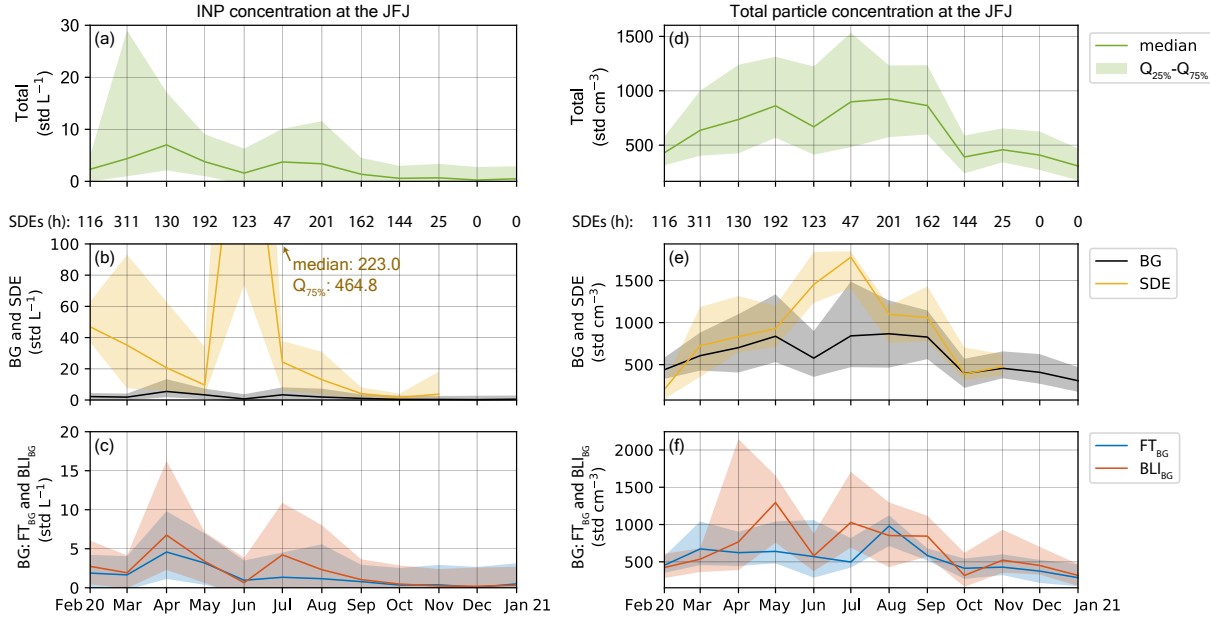

**Figure 2.** Monthly median and $25^{th}$ to $75^{th}$ percentiles ($Q_{25\%}$ - $Q_{75\%}$) of total INP concentrations (**a**), periods with Saharan Dust Events (SDE) or background (BG) conditions (**b**), and INP concentrations measured in free tropospheric background (FT$_{BG}$) or boundary layer intruded background (BLI$_{BG}$) air masses (**c**), and analogous for the total particle concentrations (**d**, **e**, and **f**, respectively) at the JFJ between February 2020 and January 2021. The number above (**b**) and (**e**) indicate the total hours of SDEs during each month. There were no SDEs in December 2020 and January 2021.

Multiple species of pollen and subpollen particles were found to be ice-active in previous laboratory studies, specifically birch, juniper, pine, orchard grass and redtop grass (e.g., Gute and Abbatt, 2020, and references in Table 1). Note, juniper
belongs to the cypress family. Given their seasonality, could pollen or subpollen particles be responsible for the distinctively higher INP concentrations in April? Figure 3 (and Fig. A1 in more detail) shows the pollen concentration for two stations closest to the JFJ, Bern (60 km northeast, 540 m a.s.l.) and Visp (29 km south-southeast, 658 m a.s.l.), however, both of them are ground sites within the PBL. Pollen were sampled by MeteoSwiss on traps for one week on a rotating drum providing a daily data resolution (see Galán et al., 2014, for more information). We assess Bern to be representative for the pollen load in
the Swiss plateau, which is directly north of the JFJ, while Visp is located in the Rhône-valley south of the JFJ. To compare, Figure 3b illustrates the INP concentrations in BG air masses (without SDEs) and the $Q_{10\%}$ of $P_{FT}$ of a given day. We chose the $Q_{10\%}$ as proxy, as brief BLIs are ignored while longer periods of BLIs are accounted for. BLIs need to prevail cumulatively for 2.5 hours a day to show a BLI signal in $Q_{10\%}$. In the following, we discuss the median freezing temperature of the pollen wash water, containing the subpollen particles of each species, where D01 denotes Diehl et al. (2001), vB05 denotes von Blohn
et al. (2005), P12 denotes Pummer et al. (2012), D17 denotes Dreischmeier et al. (2017), and G20 denotes Gute and Abbatt (2020).





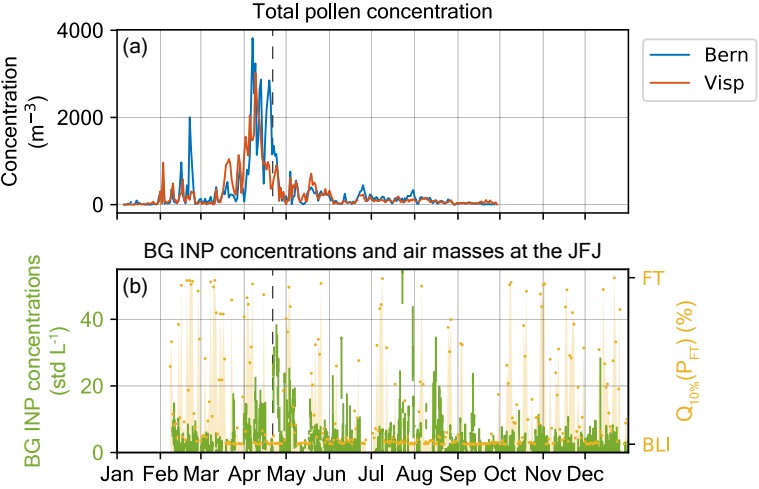

**Figure 3.** Timeline of the total pollen concentration at Bern and Visp (**a**), and the corresponding BG (= excl. SDE) INP concentration (high-resolution) at the JFJ with the $Q_{10\%}$ of $P_{FT}$ of a given day (**b**) between February 2020 and the end of December 2021. Pollen data are only available between early January until the end of September. Pollen data: courtesy of MeteoSwiss.

Pollen in the ambient air were detected between January 7 and the end of the sampling period on September 29, with a first climax at the end of February, where yew (vB05: 250 K) dominated in Bern, but was almost inexistent at Visp. The remaining major fraction stemmed from alder (D17: 256 K, G20: 264.75 K) and hazel (P12: 249 K). A second maxima, resembling the absolute climax in pollen concentration, was between April 5 and 10. Between the end of March and mid-April, the overall

pollen load in Visp was 12% higher than in Bern. It was dominated by birch (D01: 258 K, P12: 254 K, G20: 259 K) by 58% and 49% for Bern and Visp, respectively, followed by esch (Bern: 17%, Visp: 24%), poplar, and cypress (P12: 253 K, G20: 255 K). In Bern, beech, oak (D01: 256 K, G20: 259 K), spruce, and sycamore (G20: 266 K) were mainly detected between mid-April and the end of April, while in Visp the overall pollen concentration was much lower and oak was the major constituent. Birch

pollen were only found in small concentrations after April 20 in Visp and April 22 in Bern. Between June and September the dominating species were nettle (P12: 248 K) and hemp plants. All pollen, for which the ice nucleation activity was available in the literature, are active at the sampling temperature used in this study (243 K). Therefore, their subpollen particles should be detected by HINC-Auto if transported to the JFJ. Intact pollen grains itself are above the upper size cut-off of $d > 2.5$ μm of HINC-Auto (see the Brunner et al. (2021, in review) for a detailed description about the size-dependent particle transmission

efficiency). Median BG INP concentrations during the first pollen climax on February 20 - 25 were $1.4 \pm 1.2$ INP std $L^{-1}$, statistically indifferent with regard to INP concentrations 10 days before and after this period. However, the JFJ was mostly in the FT, rendering the transport of pollen or subpollen particles to the site unlikely. During the second climax on April 4 - 11, when birch pollen were dominating, median BG INP concentrations were $3.1 \pm 1.3$ INP std $L^{-1}$, and decreased to $2.4 \pm 1.2$ INP std $L^{-1}$ on April 15 - 20. The JFJ was exposed to BLI for most of this period. Between April 21 - 25, when pollen

concentrations already mostly declined, the highest median INP concentrations were measured ($20 \pm 1.4$ INP std $L^{-1}$, see also Fig. A1). Based on these observations, it is unlikely that fragmented pollen or subpollen particles are responsible for the


observed high BG INP concentrations in April, however, because no pollen measurements were available for the JFJ, this needs to be verified in future work. The background-corrected maximum contribution of fragmented pollen or subpollen particles to the overall INP population at $T$ = 243.15 K and $S_w$ = 1.04 during peak periods (April 8 - 14) is 19.9 INP std L$^{-1}$ , with a
median contribution of 5.8 INP std L$^{-1}$ during these peak periods (see also Fig. A1). If every pollen grain would be ice-active at 243 K, and the same pollen concentration were present at the JFJ as measured in Bern, i.e., the PBL was perfectly mixed and the JFJ was within the PBL, pollen would only contribute up to 3.6 INP std L$^{-1}$ (4 INP L$^{-1}$), 5 times less than the Q$_{95\%}$ INP concentration for BLI$_{BG}$ conditions during the same time period. This indicates that if the peaks of up to 19.9 INP std L$^{-1}$ result from pollen particles, a fragmentation process needs to be involved, like the swelling of pollen, burst, and subsequent
release of a great number of subpollen particles from every pollen grain, as predicted by laboratory experiments (Gute and Abbatt, 2020).

As with every instrument, HINC-Auto introduces artefacts based on the methodology used which reflect back in the data it provides. Because the measured INP concentrations are often close to or within the counting uncertainty, a simple model of the sampling method used in HINC-Auto was developed to assess the overall and seasonal BG INP concentration. In the model,
all particle-free air measurements between February 7, 2020 and January 31, 2021 (N = 21614, 5 min each) were concatenated to a single vector. Using random 5-minutes samples from this vector (with replacement), 10000 artificial measurements were simulated with a prescribed, constant INP concentration. The frequency distributions of these synthetic measurements for prescribed, constant INP concentrations of 0, 1, 2, 4, 8, and 16 std L$^{-1}$ are shown in Figure 4, and compared to the seasonal and overall INP concentrations. For a prescribed concentration of 0 INP std L$^{-1}$, the model estimates that HINC-Auto would
record values of up to 10 INP std L$^{-1}$, with a median and average close to 0 INP std L$^{-1}$. The difference between the median and the average concentration corresponds to statistical noise and converges to 0 INP std L$^{-1}$ for an increasing number of artificial measurements. 71.7% of all artificial measurements are below the LOD, which deviates from the 84.1%, being the fraction expected for values following a symmetric Gaussian distribution without a kurtosis. For a prescribed concentration of 1 INP std L$^{-1}$, the shown distribution using a log-scale does only change insignificantly compared to a prescribed concentration
of 0 INP std L$^{-1}$. The median and mean converge close to 1 INP std L$^{-1}$, while the fraction below the LOD decreases. Also for artificial measurements with higher prescribed INP concentrations, the median and mean agree well with the set prescribed INP concentration, however, the shape of the distribution follows a normal and not a log-normal distribution. The seasonal and overall frequency distributions, in contrast, show a log-normal distribution, as expected by theory (Ott, 1990) and discussed more in Brunner et al. (2021, in review). Comparing the artificial signals with the measurements, the total
(SDE+BG) true median INP concentration in winter was likely between 0 and 1 INP std L$^{-1}$. This is supported by the median observed in the INP measurements (0.7 INP std L$^{-1}$), the fraction below the LOD as well as the shape of the distribution. The INP concentration in spring increased (median: 3.1 INP std L$^{-1}$), with and without consideration of measurements during SDEs, and decreased in summer (median: 1.6 INP std L$^{-1}$) and decreased further in fall (median: 0.5 INP std L$^{-1}$). The total (including SDE and BG periods) median INP concentration between February 2020 and January 2021 is 1.4 INP std
L$^{-1}$. The analysis with a model mimicking the instrument's methodology nicely shows how an instrument with given counting uncertainty can alter the frequency distributing of INP concentrations, whereas the median remains robust. Thus, comparing the





observed seasonal and total frequency distributions with the model output with known INP concentrations, e.g., with respect to the fraction of INP concentrations below the LOD and the relative location of the median to the frequency bins above 0.01 INP std L$^{-1}$, the model supports the reported median INP concentrations. Yet, it questions which fraction of the observed frequency
distribution arises only because of the instrument bias, and subsequently, does not correspond to the true INP concentration within the atmosphere. This mainly concerns reported individual INP concentrations below 10 INP std L$^{-1}$, whereas for higher INP concentrations, the reported frequency distributions are robust, as the model shows less spread the higher the prescribed INP concentration.

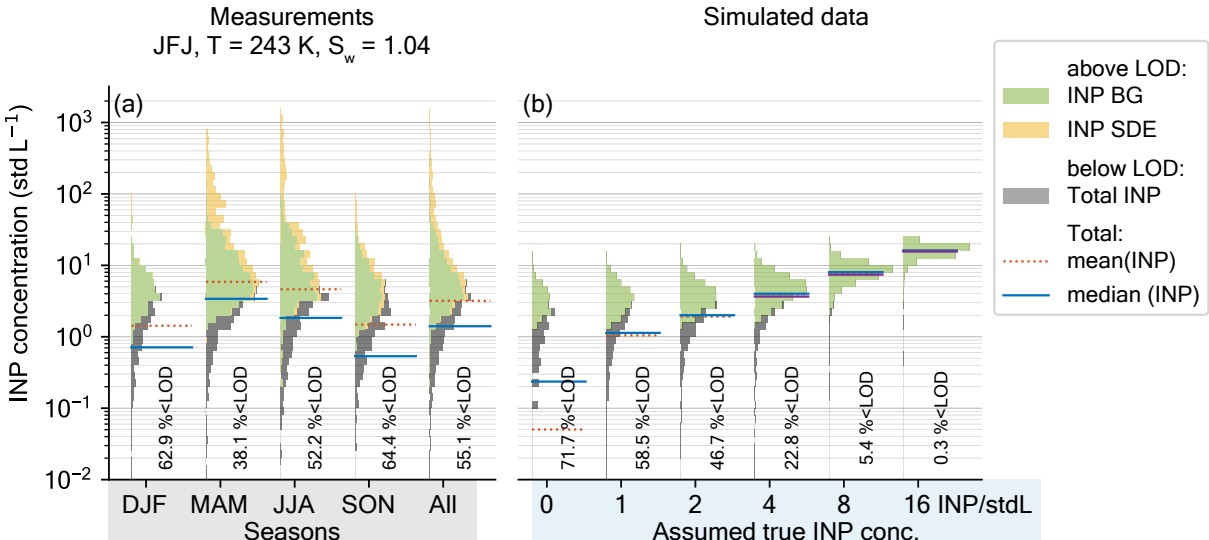

**Figure 4.** (**a**) Frequency distributions of INP concentrations measured in winter (December 2020, January 2021, February 2020), spring (March - May 2020), summer (June - August 2020), fall (September - November 2020), and during the entire investigated period (February 2020 - January 2021); (**b**) frequency distributions of simulated artificial INP concentrations with a prescribed INP concentration of 0, 1, 2, 4, 8, and 16 std L$^{-1}$. Green indicates INP concentrations above the limit of detection (LOD) during BG conditions, yellow during SDE periods, grey INP concentrations below the LOD (independently of the air mass), red dashed lines indicate the overall mean INP concentration, and the blue line the overall median INP concentration. Because of the log-scale, only positive concentrations are shown. The stated percentages indicate the fraction of measurements below the LOD.

Disentangling the total observed INP concentration for the entire period in more detail, the median BG INP concentrations
remains at 1.4 INP std L$^{-1}$, hinting to the small contribution of SDE periods to the median INP concentration, given the comparatively short total duration of all SDE periods (55 days 20 hours) within the entire period analyzed (359 days). A further division of all BG INP concentrations into periods with FT or BLI air masses reveals the median INP concentrations to be 20% lower in the FT (FT$_{BG}$ = 1.2 INP std L$^{-1}$) and 7% higher during BLI (BLI$_{BG}$ = 1.5 INP std L$^{-1}$).

Table 1 provides a more detailed overview of the seasons, including the $25^{th}$, $75^{th}$ and $95^{th}$ percentiles. In general, the
concentrations found in the present work are lower compared to earlier work at the JFJ (Lacher et al., 2018), but consistent with studies at different locations (e.g., Schrod et al., 2020). Comparing the frequency distribution of the measurements from





this work to Lacher et al. (2018) emphasises the difference in observed concentrations (see Fig. A2), and proves that the nine single field campaigns in the mentioned earlier work, all targeted to sample SDEs between mid-January and the beginning of March and between May and August in the years 2014-2017, were successful in probing an over-proportional fraction of SDEs.

This is further supported by the INP frequency distributions, which for Lacher et al. (2018) are not log-normal as expected compared to those of the present work. Overall, the seasonality has a minor impact on the observed INP concentrations, which is consistent with other work (e.g., Tobo et al., 2020; Schrod et al., 2020). This statement holds for FT conditions, was well as within the PBL. The seasonal INP number concentrations vary by a factor of up to 7 for identical statistical metrics, e.g., when comparing the median concentrations, thus, well below the variation observed within all measurements or compared to SDEs.

**Table 1.** Seasonal statistics of INP concentrations measured at the JFJ at $T = 243.15$ K and $S_\mathrm{w} = 1.04$ between February 7, February 7, 2020 and January 31, 2021.

| INP concentration $\pm \sigma$ (std L$^{-1}$) | $Q_{25\%}(INP)$ | Median(INP) | $Q_{75\%}(INP)$ | $Q_{95\%}(INP)$ | $P_{FT}$ |
|---|---|---|---|---|---|
| BG: | | | | | |
| Overall: | -0.9 ± 1.0 | 1.4 ± 1.0 | 4.7 ± 1.2 | 15.1 ± 1.6 | 40.4% |
| FT$_{BG}$: | -1.0 ± 1.0 | 1.2 ± 1.0 | 4.2 ± 1.1 | 11.7 ± 1.4 | 100.0% |
| BLI$_{BG}$: | -0.9 ± 1.5 | 1.5 ± 1.1 | 5.1 ± 1.4 | 18.1 ± 1.6 | 0.0% |
| Spring (MAM): | 0.5 ± 1.1 | 3.4 ± 1.2 | 7.8 ± 1.1 | 21.4 ± 1.4 | 48.4% |
| Summer (JJA): | -0.7 ± 0.8 | 1.8 ± 1.1 | 6.3 ± 1.4 | 22.3 ± 1.6 | 20.4% |
| Fall (SON): | -1.0 ± 1.1 | 0.5 ± 1.1 | 2.9 ± 1.1 | 9.7 ± 1.4 | 42.3% |
| Winter (DJF): | -1.1 ± 0.9 | 0.7 ± 1.2 | 3.4 ± 1.2 | 8.6 ± 1.2 | 47.3% |
| SDE: | | | | | |
| FT$_{SDE}$ | 7.4 ± 1.1 | 17.3 ± 1.1 | 46.1 ± 1.3 | 112.7 ± 1.3 | 100.0% |
| BLI$_{SDE}$ | 7.4 ± 1.2 | 23.7 ± 1.5 | 74.6 ± 1.5 | 354.9 ± 1.5 | 0.% |

## 3.2 The diurnal INP variability at the JFJ

To study the diurnal variability of INPs at the JFJ, phase-statistics of the BG INP concentrations with a cycle period of 24 h were calculated for the total investigated period, starting at midnight of each day. These phase-statistics were divided into periods when the full day was in FT air masses (30 days), days not exclusively in FT or BLI (244 days), or days only with BLI (91 days), as shown in Figure 5 with statistics in Table 2. Phase-statistics of the total particle number concentrations during

BG periods are superimposed (total particle concentrations from CPC measurements without SDE periods). For days only in the FT, there is no clear diurnal cycle in INP number concentrations evident, while the median BG particle concentration





**Table 2.** Diurnal statistics of BG INP concentrations at $T = 243.15$ K and $S_w = 1.04$ and BG total particle number concentrations ($\leq 14$ nm), for both the median of the phase-averaged concentrations measured at the JFJ between February 7, February 7, 2020 and January 31, 2021.

| $Q_{50\%}$ INP concentration (std L$^{-1}$) | Maximum | Time of maximum (UTC) | Minimum | Time of minimum (UTC) | No. days |
|---|---|---|---|---|---|
| BG: | | | | | |
| Days only in FT$_{BG}$: | 2.1 | 11:45 h | 0.26 | 19:00 h | 30 |
| Days not exclusively in FT$_{BG}$ and BLI$_{BG}$: | 1.7 | 15:00 h | 0.74 | 6:00 h | 244 |
| Days only with BLI$_{BG}$: | 2.1 | 16:30 h | 0.64 | 2:00 h | 91 |
| $Q_{50\%}$ total particle number concentration (std cm$^{-3}$) | | | | | |
| BG: | | | | | |
| Days only in FT$_{BG}$: | 629 | 13:00 h | 330 | 8:30 h | 30 |
| Days not exclusively in FT$_{BG}$ and BLI$_{BG}$: | 824 | 15:00 h | 506 | 6:00 h | 244 |
| Days only with BLI$_{BG}$: | 1263 | 14:30 h | 732 | 3:30 h | 91 |

shows a weak diurnal cycle, with a maximum of 629 std cm$^{-3}$ at 13 h UTC and a minimum of 330 std cm$^{-3}$ at 8:30 h UTC. For days with a mix of FT and BLI air masses, the median BG INP and BG particle number concentrations follow a similar diurnal cycle, with maximum of 1.7 std L$^{-1}$ and 824 std cm$^{-3}$, respectively, at 15 h UTC, and a minimum of 0.74

std L$^{-1}$ and 506 std cm$^{-3}$, respectively, at 6 h UTC. The variation is less pronounced in BG particle concentrations (max/min vs. cycle median = +39%/-14%) compared to the observed variation in BG INP concentrations (max/min vs. cycle median = +70%/-26%). Also for days entirely with BLI a diurnal variability in BG INP concentrations is apparent. The mean BG particle concentrations increase by 58% compared to mixed days, and double compared to days entirely in the FT, while median BG INP concentrations increase by 33%, and 32% for pure BLI days compared to mixed, and FT days, respectively. The maximum

BG INP and BG particle number concentration is at 16:30 h UTC (2.1 std L$^{-1}$) and 14:30 h UTC (1263 std cm$^{-3}$), while the minimum was at 2 h UTC (0.64 std L$^{-1}$) and 3:30 h UTC (732 std cm$^{-3}$). As for days with FT and BLI, the relative variation for BG INP concentrations is more pronounced. The results suggest that the diurnal cycle of INPs is driven by convection and the expansion of the planetary boundary layer throughout the day, as discussed below.





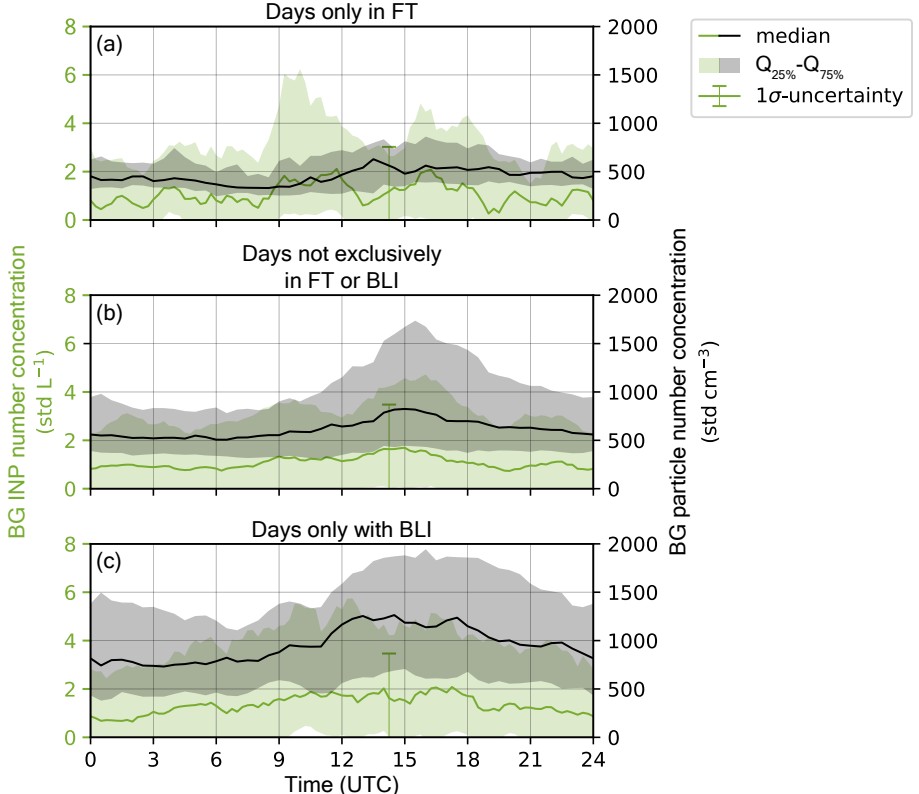

**Figure 5.** Median and quartiles of the BG INP concentrations (green) and BG particle number concentrations (grey), phase-averaged with a cycle period of 24 h, starting at 0 h UTC of each day between February 2020 and January 2021, for measurements from days only within the FT (**a**), for days not exclusively in FT or BLI (**b**), and for days only with BLI (**c**). Error bars show a $\sigma$ counting uncertainty of the INP measurements.

Figure 6 illustrates a case study of the diurnal cycle during a convective period between March 22 - April 15, excluding SDE periods. On March 22, an anticyclone with its center over Sweden extended its influence to central Europe with prolonged fair weather. Between March 25 and 30, a SDE was recorded at the JFJ. On March 30, an occluded front passed the Alps from northwest within 24 h, followed by an increase in surface pressure. An anticyclone moved across Europe and shielded the Alps until April 16 from further frontal passages, however, on April 6 and 7, a second weak SDE was recorded at the JFJ. For the period between March 22 - April 15, the INP concentration shows a diurnal cycle, with a median INP concentration between 4 and 5 std L$^{-1}$ throughout the night and peaking at 16 h UTC up to 14 std L$^{-1}$. The Q$_{75\%}$ concentrations peak earlier in the day, at noon with 35 std L$^{-1}$. Compared to the total particle concentration, the median diurnal INP cycle is again less pronounced and declines sooner after peaking. $^{222}$Rn concentrations, a tracer for boundary layer intrusions, resemble the diurnal cycle in total particle concentrations well, and decline later after peaking compared to the median INP concentrations. In addition, both total particle concentrations and $^{222}$Rn concentrations decline steadily until the morning, while the INP concentrations remain fairly constant throughout the night. The ceilometer at Kleine Scheidegg reports an increased attenuated backscatter





signal in the afternoon for the same altitude as the JFJ (3580 m a.s.l.) , indicating the rise of the PBL height and the formation of isolated convective clouds (see Fig. 6d). The signal drops quickly after peaking at 16 h UTC. Thus, it resembles more the trend seen in the INP concentrations than the one of the total particle concentrations or $^{222}$Rn concentrations. Concentrations of particles with a diameter larger than 0.5 and 2.0 μm ($N_{0.5\mu m}$ and $N_{2.0\mu m}$, respectively) are often used as a predictor of INP

concentrations, yet, they show a different diurnal cycle than the INP concentrations. $N_{0.5\mu m}$ shows after a first peak at 16 h UTC a drop of more than 50%, followed by a second peak at 22 h UTC. Also $N_{2.0\mu m}$ shows two peaks, one at 16 h UTC with 106 std $L^{-1}$ and a second, higher peak at 18:30 h UTC. The large particle concentrations continue to decrease between 9 - 12 h UTC, which does not correspond to the night-time trend seen in the INP concentrations. The large fluctuations in $N_{0.5\mu m}$ and $N_{2.0\mu m}$ are unexpected, but the quartiles indicate the signals to be robust and data is available without any gaps for the full

period investigated. Overall, this strengthens the suggestion that BG INP concentrations follow the convective diurnal cycle at the JFJ, hence implying BLI to be a relevant source of INP at the JFJ during non-SDE periods.



**Figure 6.** Median and quartiles of the INP concentrations (**a**), total particle concentration (**b**), $^{222}$Rn decay rates (**c**), the attenuated backscatter signal at 3580 m a.s.l. of the ceilometer at the Kleine Scheidegg (KSE, **d**), particle concentration with a diameter larger than 0.5 μm (**e**), and particle concentration with a diameter larger than 2.0 μm (**f**). All data are phase-averaged with a cycle period of 24 h, starting at 0 h UTC of each day between March 22 and April 15, without the two SDEs (March 25 - 30, and April 6 - 7). All measurements except for the ceilometer are performed at the JFJ.



## 4 Conclusions

Continuous, sub-hourly measurements of the ambient INP concentration enable to statistically study the behavior of INPs during repeating meteorological events. Such long-term measurements were absent so far. In this work, continuous online INP measurements at $T$ = 243.15 K and $S_\mathrm{w}$ = 1.04 at the JFJ between February 7, 2020 and January 31, 2021 were analyzed with regard to their seasonality and diurnal cycle. We found a seasonal cycle, highest in spring with a median of 3.1 INP std L$^{-1}$, followed by summer (median: 1.6 INP std L$^{-1}$), and lowest in fall and winter (median: 0.5 std L$^{-1}$ and 0.7 INP std L$^{-1}$, respectively), all in absence of SDEs. This is consistent with the seasonality observed in other studies at the same site (Conen et al., 2015; Lacher et al., 2018) and at different sites (Wex et al., 2019; Tobo et al., 2020; Schneider et al., 2020). Here, INP concentrations show a larger seasonal dependency than the total particle concentrations. We hypothesize this to be an effect of the different seasonality of the partitioning types of particles, e.g., mineral dust for INPs at 243 K vs. biological and anthropogenic particles for the total aerosol concentration. A positive correlation between ambient temperature and the INP concentrations was non-existent, in contrast to earlier studies for INPs active at warmer temperatures. Based on our observations, it is unlikely that pollen or subpollen particles are responsible for the observed high background INP concentrations in April, although during peak periods their contribution cannot be ruled out up to 19.9 INP std L$^{-1}$ to the overall INP population at $T$ = 243.15 K and $S_\mathrm{w}$ = 1.04. The seasonal quartile concentrations vary by a factor of up to 7 for identical statistical metrics, which is much smaller than the observed variation due to special events, e.g., SDEs, which can cover three orders of magnitudes. No diurnal INP cycle was found for days purely in the FT, indicating that sinks and sources of INPs in the FT are either far away from the JFJ or do not follow a diurnal pattern. Atmospheric ageing, for example, which potentially makes atmospheric particles ice-active, is either a slow, (ultra violet-) light dependent process or happens only on local scale. For days with a mix of FT and BLI or for days entirely with BLI, a diurnal cycle similar to the diurnal cycle in total particle concentration was found, yet more pronounced in the case of the diurnal INP variability. Limitations were faced concerning the counting uncertainty of the instrument in combination with the low ambient INP concentrations, hindering the study of INP concentrations at warmer temperatures. By using an aerosol concentrator, future measurements can be extended to warmer sampling temperatures, e.g., for INPs active at 248 K. While this study covered almost a full year, future studies over multiple years to decades can help to fill knowledge gaps in spatiotemporal variability of INPs. The investigation of the interannual variability and trends could, for example, provide some insight whether the observed seasonality is linked to other parameters or how the anthropogenic land-use change and desertification (Ginoux et al., 2012) affect the INP number concentrations in the atmosphere.

*Data availability.* The data presented in this publication will be made available at DOI: xx.xxxx/ethz. Note by authors: data will be made available upon publication





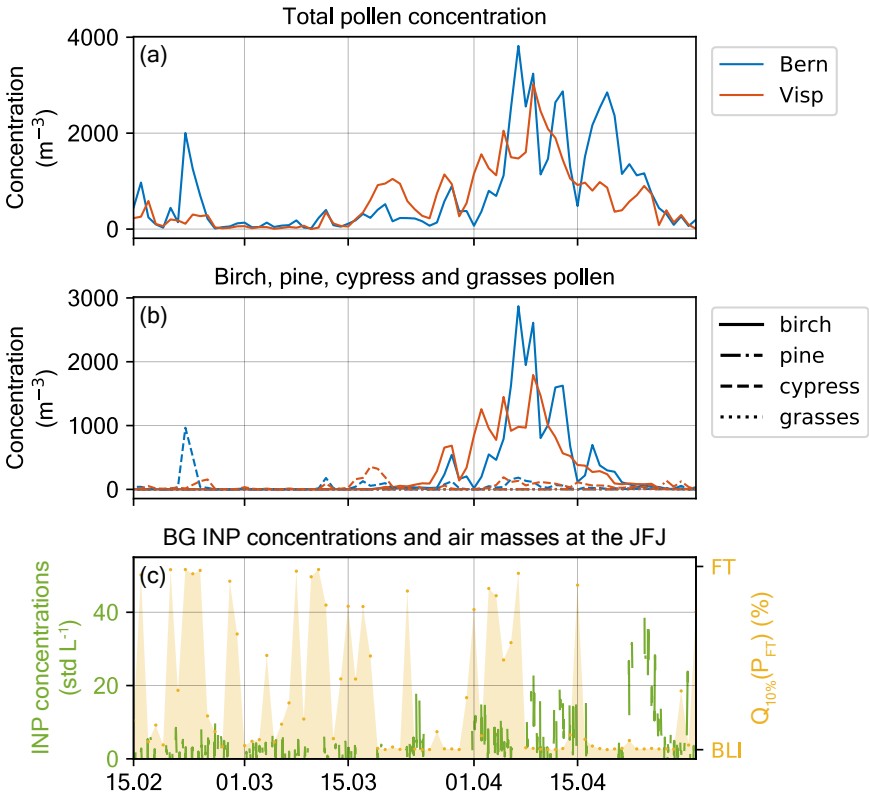

**Figure A1.** Timeline of the total pollen concentration at Bern and Visp (**a**), of the dominant species (**b**) and the corresponding high-resolution INP concentration at the JFJ and the $Q_{10\%}$ of $P_{FT}$ of a given day (**c**) between February 15 and May 1, 2020. Pollen data, courtesy of MeteoSwiss.





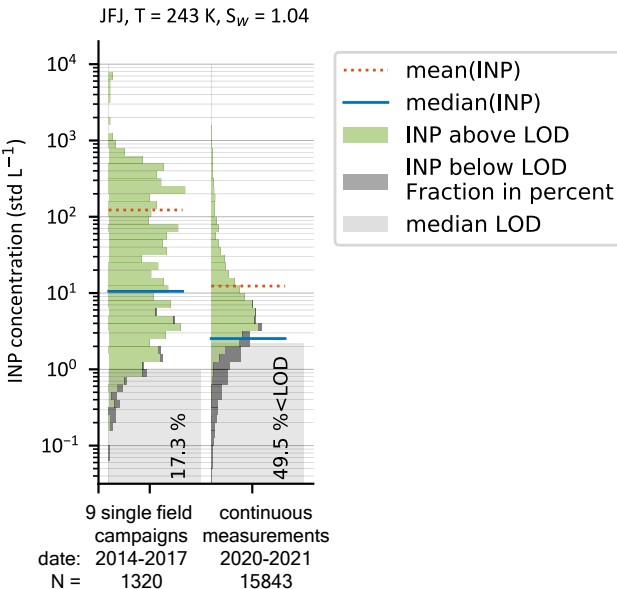

**Figure A2.** Total INP frequency distributions from 9 single field campaigns between 2014 and 2017 by Boose et al. (2016) and Lacher et al. (2018) and continuous measurements from the curent work between February 2020 and January 2021, both sampled at the JFJ including SDE and BG periods.

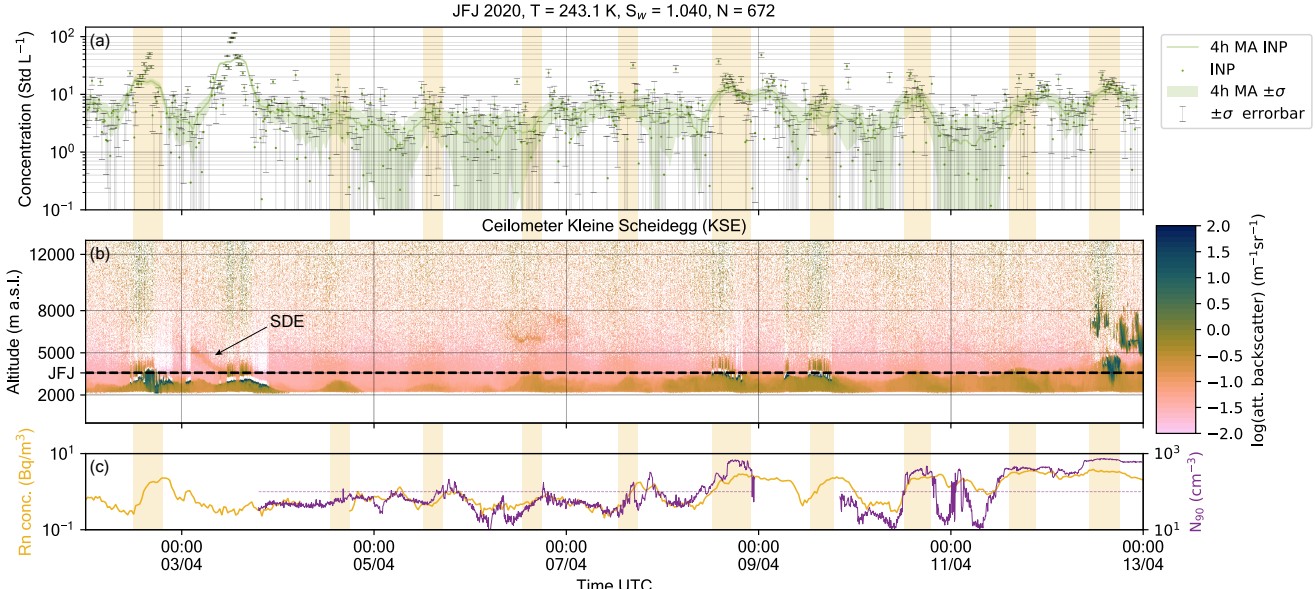

**Figure A3.** INP concentrations at the JFJ, including a 4 hour moving average (MA) and a 1 $\sigma$ counting uncertainty (**a**), attenuated backscatter from the ceilometer at the Kleine Scheidegg (**b**), and Radon and $N_{90}$ concentrations (**c**) during a convective period between April 2 and 13, 2020. Note the peaks in INP concentrations coincide with the diurnal swelling of boundary layer air masses past the altitude of the JFJ, highlighted by the orange shading and indicated by the increase in backscatter from $\leq 10^{-0.8}$ to $\geq 10^{-0.5}$ m$^{-1}$ sr$^{-1}$.

*Author contributions.* CB wrote the manuscript with input from BB, FC, MC, MS, MGB, and ZAK. ZAK conceived the field study. CB conducted all INP measurements and analyzed all INP, CAMS and ceilometer data. CB and ZAK interpreted the INP data and prepared the figures. BB and MGB contributed data aerosol particle concentrations, $N_{90}$, absorption and scattering coefficients. MC contributed data on the single scattering albedo. FC contributed data on the Radon concentration and developed the air mass classification (FT or BLI). MS contributed data on trace gases and PM. ZAK supervised the project and obtained funding.

*Competing interests.* The authors declare that they have no conflict of interest.

*Acknowledgements.* The authors gratefully acknowledge the Centre for Environmental Data Analysis (CEDA) for providing the ceilometer data, more specifically Maxime Hervo from MeteoSwiss, the Deutscher Wetterdienst, and Olivier Trollé from MétéoFrance, MeteoSwiss for the pollen data, the Swiss contribution to ICOS (https://www.icos-ri.eu) for supporting the operation of the radon detector and the operation of the ceilometer at Kleine Scheidegg, and the Copernicus Atmosphere Monitoring Service for provding the CAMS data. In this work CAMS data is generated using Copernicus Atmosphere Monitoring Service information (2021). It is important to note, that neither the European Commission nor ECMWF is responsible for any use that may be made of the Copernicus information or data it contains. We





also thank Dr. Stephan Henne from the Swiss Federal Laboratories for Materials Science and Technology (Empa) for providing FLEXPART

simulated particle surface residence times. This research was funded by the Global Atmospheric Watch, Switzerland (MeteoSwiss GAW-CH+ 2018–2021). We acknowledge that the International Foundation High Altitude Research Stations Jungfraujoch and Gornergrat (HFSJG), 3012 Bern, Switzerland, which made it possible for us to carry out our experiment(s) at the High Altitude Research Station at Jungfraujoch, with a special thanks to Claudine Frieden, Prof. Dr. Markus Leuenberger and the custodians Joan and Martin Fischer, Christine and Ruedi Käser, and Daniela Bissig and Erich Furrer. The radon observations at Jungfruajoch and the ceilometer observations at Kleine Scheidegg

are supported by the Swiss National Science Foundation (SNSF) as a contribution to the pan-European Integrated Carbon Observation System (ICOS) Research Infrastructure. The continuous aerosol measurements at the Jungfraujoch site are supported by MeteoSwiss in the framework of the Swiss contributions (GAW-CH) to the Global Atmosphere Watch program of the World Meteorological Organization (WMO), and the ACTRIS research infrastructure funded by the Swiss State Secretariat for Education, Research and Innovation (SERI) and by the European Commission under the Horizon 2020 - Research and Innovation Framework Programme, H2020-INFRADEV-2019-2, Grant

Agreement number: 871115 (ACTRIS IMP). We thank Prof. Dr. Ulrike Lohmann for her support and enthusiasm. We acknowledge Dr. Heike Wex, Jörg Wieder, Dr. Zane Dedekind, Dr. Larissa Lacher, Dr. Fabian Mahrt, Julie Pasquier, and Dr. Carolin Rösch for useful discussions. For technical support and fabrication, we would like to thank Dr. Michael Rösch and Marco Vecellio, whose expertise greatly helped to improve the instrumentation.



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
