# Peer review of "The diurnal and seasonal variability of ice nucleating particles at the High Altitude Station Jungfraujoch (3580 m a.s.l.), Switzerland"

_Atmospheric Chemistry and Physics, 2021_

## Author Comment (AC1)

**Response to RC1**

We have reproduced reviewer comments in **bold** font, and our responses in regular font.

**General comments**
**The manuscript investigates the seasonal and diurnal variability in ice nucleating particle oncentration (INP) measured over a year at the High Altitude Research Station Jungfraujoch. It represents the longest continuous measurement of INPs to date with a high time resolution of 20 minutes. A easonal trend in INPs is observed with highest concentrations occurring in Spring and lowest concentrations occurring in Winter. A diurnal trend in INPs is also identified for air masses with boundary layer intrusions. The study identifies long term trends in INP concentrations and is a valuable contribution to the field of INP research. However, I believe that the discussion of the data represented in the figures could be clearer and further links between potential INP sources and with previous literature studies could be made. I therefore recommend the manuscript for publication in ACP following appropriate response to the following comments.**
We thank Reviewer 1 for their positive evaluation of the manuscript, and respond to their comments individually below with corresponding modifications to the manuscript indicated by the respective line numbers in the revised version.

**Specific comments**
**1. The results section contains very detailed analysis and lots of information is contained within each figure. It would be clearer and easier for the reader to follow the discussion if the panel or the section of the figure that is being discussed is regularly referred to in the text.**
**e.g. for Figure 2:**
- **page 8, line 204: 'Dividing BG periods into FTBG and BLIBG...' Please refer to panel c).**
- **Page 8, line 216: '...is apparent in April for the total particle concentration.' Please refer to panel f).**
- **Page 8, line 217: '...total particle number concentrations remained at summer levels also in September' Please refer to panel d).**

We agree, and made the above recommended changes to refer to Fig. 2 (see lines 203, 213, 215, and 217 resp. in revised version)

**This comment applies to all figures but especially to figures 2, 5 and 6.**
We agree, and have added references to Fig. 5 in the text (see lines 329, 331, 333, 335, 337, 339 - 341, 344 in revised manuscript), Figure 6 (lines 352, 354-355, 359-360, 363) and for Figure 2 we modified the references based on the specific comments of reviewer 1 above.

**Additionally it would also help the reader if colours were referred to in the text when discussing the data, e.g. in Figure 5, page 14, line 322-323: '...shows a weak diurnal cycle, with a maximum of 629 std cm-3 at 13 h UTC and a ...' Please add (black line in panel a)).**
We have now made references to the colour of the lines and traces as well in the revised manuscript when we refer to the specific figure panels (see response to comment above).

**2. Pages 9-11 contains a detailed discussion of pollen as the potential INP source for the high INP concentrations measured in April. Whilst this discussion is interesting, I believe it could be reduced as**

**the overall conclusion is that it is unlikely that pollen is responsible for the high INP concentrations in April (without further pollen measurements at JFJ). Why do you not comment on any other potential sources for the high INP concentration in April? Was any back trajectory analysis of air masses performed that could inform on potential INP sources?**

We believe the discussion on pages 9-11 is necessary to conclude that we cannot rule out the contribution of sub-pollen particles or pollen fragments to the peak INP concentrations observed in April, which we now make explicit (see lines 264 – 277 in revised manuscript). We now specify that we only rule out the influence of pollen grains because the HINC-auto inlet had a $D_{50}$ size cut-off of 2.5 µm (see lines 248 initial manuscript and line 256 revised manuscript) which does not allow sampling of pollen grains. We have now made it more clear in the conclusions section that the role of sub-pollen particles and pollen fragments cannot be ruled out (see lines 391 – 393 in revised manuscript).

We have conducted a back trajectory analysis (see Fig. 1 below) for and exemplary time during the peak INP concentrations. Fig. 1 below shows the back trajectory analysis initiated for April 22nd at 1200 hrs UTC. The analysis shows that the air masses with recent ground contact were advected from either side of the main Alpine ridge, i.e., from both north or the south of JFJ, thus it is possible that sub-pollen particles could be transported to the JFJ from the south (Italy) or other regions of Switzerland. However, it is clear that trajectories do not arrive from Bern. Trajectories could have arrived at JFJ on April 22nd from the region near Visp, however, only after April 19, when the pollen concentrations had already declined (i.e. after April 11-15). Thus, we do not believe these two locations would contribute to the INP concentration peak observed (see Figure 3 in revised manuscript) during April 20-25th. The sources of sub-pollen particles could have been from elsewhere in Europe as also suggested by RC2. We now add this modification in lines 266-270 of the revised manuscript.

**Were any samples collected (gas or filters) and analysed for chemical composition?**
No filters or gas samples were collected that matched the time resolution of HINC-Auto for this period or analysed. Filter chemical composition could provide insight, but the resolution is on the order 24-hours, whereas the analysis here is obtained by having the higher 20 min time resolution. However, after a longer-term data set has been collected by HINC-Auto, the PM filter composition with resolution of 24-hrs could provide beneficial knowledge for future studies.

[Figure]

*Figure 1. Back trajectories(HYSPLIT, NOAA) [Rolph et al., 2017; Stein et al., 2015] ending at JFJ at 12.00 UTC on April 22nd showing no air masses arrive from Bern and Air masses arriving from Visp would only have passed over Visp after April 19, when pollen concentrations in Visp had already declined. The colours represent different trajectories starting every 3 hours going backwards. ([https://www.ready.noaa.gov/HYSPLIT_traj.php](https://www.ready.noaa.gov/HYSPLIT_traj.php)).*

**3. The introduction discusses trends in seasonal and diurnal variability in INP measurements in the literature from various studies using mostly offline analysis. It would be good to make links back to the findings of these studies during the results section for comparison i.e. similar seasonal dependences were observed.**

We have now added a paragraph in section 3.1 (see lines 222-230 in revised manuscript) to compare the seasonal cycle observed in this study compared to the other studied mentioned in the introduction.

For the comparison of the diurnal INP cycle to the literature, see comment 5 below.

**4. The introduction states that knowledge of seasonal and diurnal variability will help to understand the sources and sinks of INPs. The conclusion only briefly mentions that the observed seasonal variation of INP concentrations could be linked to partitioning of particles in different seasons. As this appears to be the main motivation for the measurements, this discussion should be expanded in either the results or conclusion section.**

We have dedicated the entire section 3.1 of the manuscript to the discussion of the seasonality of INP at JFJ. Motivated by the reviewer comments we have added a statement in the conclusions elaborating further on the seasonality observations (see lines 381-389 in revised manuscript).

**Can any further information on sources and sinks of INPs at JFJ be obtained from this study?**

Recently *Lacher et al.* [2021] showed the need for a significant number of additional necessary instruments in addition to an INP counter at Jungfraujoch during a field campaign to categorically and correlationally support the presence of other sources contributing to INP populations at JFJ. Indeed these other sources included particles of marine origin, dust and anthropogenic sources. However, this required a suite of mass spectrometers and offline chemical composition measurements of ice residuals to be running in parallel. A much bigger effort with chemical composition identification during the continuous monitoring for 12 months would be necessary. As such, a detailed assessment of sources is beyond the scope of the data set of monitoring presented here. We now make a statement in the conclusions referencing these sources and a study conducted at the same site with an instrument similar to the one used here, but a non-automated version (see line 393 revised manuscript). We also add to the conclusions that majority of the particles observed as INPs at -30 °C at JFJ should be from dust (Brunner et al., 2021) (see line 396 revised manuscript)

**5. Comment: The only other study to have observed diurnal variation in INPs over a longer time period is mentioned on page 3, lines 65-69 (Wieder et al., 2021 in prep.). It would be useful to make further comparisons between this study (data in Figure 5) and that of Wieder et al., however, as the manuscript is in prep this is not possible.**

We thank the reviewer for this comment. Indeed *Wieder et al.* [now 2022] is no longer in review and instead published. A comparison between the two studies is in principle possible but needs to be caveated. The diurnal cycle in *Wieder et al.* [2022] caused by advection and orographic lifting could generally apply to the JFJ site as well. However, the meteorological transport assessed in much greater detail and extremely specific to the topography/valley system of the Davos region with data during the field campaign taken from a wide variety of observations set-up at several sites adjacent to the site observing INP concentrations. In *Wieder et al.* [2022], we measured INP concentrations at a mountaintop and a high valley site, simultaneously and comparing these two sites, allowed us to reach a conclusion on how topography-influenced meteorology impacted the diurnal cycle at the mountaintop but not in the valley.

We now mention the agreement in the conclusions section (see lines 402-404 in revised manuscript) and add a few sentences to discuss the nature of the diurnal concentration in section 3.2 (see lines 370-374 in the revised manuscript).

**Technical corrections**

**Page 7, line 196-7: the text states that 'June had the most active SDE of the investigated period with a duration of 116 h' whereas in Figure 2 it appears that the SDE in June lasts for 123 hours. Please correct.**

Thank you for the correction, now corrected (see line 196 in revised manuscript)

**Page 16, line 352: the text states 'The large particle concentrations continue to decrease between 9-12 h UTC…' which I think should be 21-24 h UTC from the data presented in Figure 6, panel f). Please correct.**

We have corrected this (see line 366 in revised manuscript)

**Figure A3 is not mentioned in any part of the paper. Is this needed?**
Thank you for catching that, the reference to Figure A3 is now made in the results section to show the annual overview time series of INP concentrations and related parameters used to assess the seasonal and diurnal variability of INP concentrations (see lines 169 – 171 in revised manuscript)

**Typing errors/grammar:**

**Page 1, line 13: '…is with a factor of…' should be changed to 'is within a factor of'.**
Corrected (line 13 revised manuscript)

**Page 3, line 81: 'Furthermore, the remote location allows to study…' should be 'Furthermore, the remote location allows the study of…'**
Corrected (line 81 revised manuscript)

**Page 4, line 98: unites should be units.**
Corrected (line 98 revised manuscript)

**Page 6, line 181: 'There were two exceptionally dry period in the end…' should be 'There were two exceptionally dry periods at the end…'**
Corrected (line 180 revised manuscript)

**Page 11, line 291: '…uncertainty can alter the frequency distributing…' should be '…uncertainty can alter the frequency distribution…'**
Corrected (line 303 revised manuscript)

References

Lacher, L., H. C. Clemen, X. Shen, S. Mertes, M. Gysel-Beer, A. Moallemi, M. Steinbacher, S. Henne, H. Saathoff, O. Möhler, K. Höhler, T. Schiebel, D. Weber, J. Schrod, J. Schneider, and Z. A. Kanji (2021), Sources and nature of ice-nucleating particles in the free troposphere at Jungfraujoch in winter 2017, *Atmos. Chem. Phys.*, *21*(22), 16925-16953, doi:10.5194/acp-21-16925-2021.

Rolph, G., A. Stein, and B. Stunder (2017), Real-time Environmental Applications and Display sYstem: READY, *Environmental Modelling & Software*, *95*, 210-228, doi:https://doi.org/10.1016/j.envsoft.2017.06.025.

Stein, A. F., R. R. Draxler, G. D. Rolph, B. J. B. Stunder, M. D. Cohen, and F. Ngan (2015), NOAA's HYSPLIT Atmospheric Transport and Dispersion Modeling System, *Bull. Amer. Meteorol. Soc.*, *96*(12), 2059-2077, doi:10.1175/BAMS-D-14-00110.1.

Wieder, J., C. Mignani, M. Schär, L. Roth, M. Sprenger, J. Henneberger, U. Lohmann, C. Brunner, and Z. A. Kanji (2022), Unveiling atmospheric transport and mixing mechanisms of ice-nucleating particles over the Alps, *Atmos. Chem. Phys.*, *22*(5), 3111-3130, doi:10.5194/acp-22-3111-2022.

---

## Author Comment (AC2)

**Response to RC2**
We have reproduced reviewer comments in **bold** font, and our responses in regular font.

**Review of "The diurnal and seasonal variability of ice nucleating particles at the High Altitude Station Jungfraujoch (3580 m a.s.l.), Switzerland" by Brunner et al.**

**The paper from Brunner et al, reports seasonal variability and diurnal variability of INP concentration at the JFJ site during the year 2020. This is this year the third paper in this series of INP measurements at JFJ. The first technical paper appeared earlier this year in AMT, describing the auto-HINC, a new CFDC device enabling continuous INP measurement at the JFJ. "Continuous online monitoring of ice-nucleating particles: development of the automated Horizontal Ice Nucleation Chamber (HINC-Auto)". A second paper, "The contribution of Saharan dust to the ice nucleating particle concentrations at the High Altitude Station Jungfraujoch (3580 m a.s.l.), Switzerland" currently in ACPD presents one year (2020) data of INP attributed to Sarahan dust and measured at JFJ. This current third paper is pushing the analysis further by looking more carefully at the seasonal variation of the INP during the same time, extracting the INP seasonality and diurnal variation by excluding the SDE. The data are first cleaned from local pollution (roughly 25% of the data removed) and then data is classified in 4 different air masses: FT with or without SDE and BLI with/without SDE. This paper is well written and very pleasant to read. It is a nice continuation of the first two papers published/under review this year. It presents an impressive work of high temporal resolution of INP concentration for 1 year of continuous measurement. The fact that the authors could use this high temporal resolution HINC instrument compared to "classical" daily filter measurement allow the authors to remove from the data any short local pollution, which I'm not sure would have been feasible with 24hr filter. For sure, much more of this type of high temporal INP measurement is very appreciated, and hopefully more in the future will be done (also at different locations).**

**I have only one main comment and a few small comments, and I recommend the paper to be published once these comments are addressed.**
We thank the reviewer for their comments and positive evaluation of the paper. We respond to the reviewer concerns individually below and indicate modifications with line numbers made in the revised manuscript.

**Main comment:**
**PL369: "Based on our observations, it is unlikely that pollen or subpollen particles are responsible for the observed high background INP concentrations in April,"Looking at the data, I would arrive at a different conclusion (or at least less affirmative about the non-influence of pollens on INP at JFJ).**
We agree, and based on Reviewer 1 comments, we have modified the specific statements to be less affirmative about the absence of pollen influence on INP concentrations in April. First, we re-iterate in the conclusions that we do not expect pollen grains to contribute to the INP concentration merely because of the $D_{50}$ size cut-off of the sampling inlet at HINC (this is already mentioned in the results section, see line 256 revised manuscript), and we now emphasize this in lines 390-393 of revised manuscript in the conclusion section. Also see comments below for back trajectory analysis and associated modifications to the manuscript.

**A) Like Sarahan dust, pollens are known to be a very good candidate to act as INP as the authors explained, and this is why the authors investigated this specific source of INP. However, the authors do not have a direct measurement of pollen directly on site, so they have to speculate. The data presented here show that there is a peak of pollen measured 60 km away from the station (at Bern, 3 km lower in altitude) just a few days (1-3 days, hard to read from the figure FA1.a) before the measured "peak" of the INP at JFJ. Pollens are released first before INP increases, which therefore**

**does not rule out the possibility of Pollen reaching JFJ and increasing INP concentration (the other way round would not work). Similar to SDE, pollen transported to JFJ could have departed days earlier before arriving on this high altitude site?**

We agree, and we have made this more clear that we cannot rule out the role of pollen fragments and sub-pollen particles contributing to the INP form other regions. Regarding Bern and Visp, since the peak in INP concentrations at JFJ was only observed around April 21-25 we ran a back trajectory analysis ending at JFJ on April 22 at 12.00 UTC on April 22 (see Fig. 1 below) with trajectories coming from the north and south with ground contact. The trajectories arrive from regions covering near Visp, but would not originate before April 15 from this region (Fig. 1 below) when pollen concentration was peaking (Fig. 3 manuscript). i.e., in Visp the pollen concentrations were already declining starting April 11-15. As such, pollen fragments from Visp will likely not contribute to the peak in INP concentrations at JFJ between April 20-25[th]. Furthermore, for air masses arriving at JFJ on April 22[nd], the trajectories do not pass over Bern which also suggests that it could not be a source of INP to JFJ during the peak on April 20-25[th]. Given a second peak in pollen was observed in Bern around April 19/20, we ran back trajectories ending at JFJ on April 20 at 12.00 UTC (see Fig. 2 below) also shows that no trajectories arriving at JFJ on the 20[th] or 22[nd] of April pass through Bern or Visp. We now acknowledge this in line lines 264-277 revised manuscript.

We further explicitly state that we cannot exclude the contribution of pollen fragments and sub pollen particles coming from other parts of Europe and Switzerland, since some trajectories came from north Italy (see Fig. 1 below) during April 22[nd] (see lines 266-268 and 276-277 revised manuscript) and the regions in Switzerland north of JFJ (see Fig. 2 below).

**B) Then there is the estimation of how many pollen particles would reach the station: P11L261 ". If every pollen grain would be ice-active at 243 K, and the same pollen concentration were present at the JFJ as measured in Bern, i.e., the PBL was perfectly mixed and the JFJ was within the PBL, pollen would only contribute up to 3.6 INP std L−1 (4 INP L−1 ), 5 times less than the Q95% INP concentration for BLIBG conditions during the same time period. "**

**However, pollen concentration measured at Bern is an average of 24hrs, whereas INP concentration measured at JFJ is a snapshot of 20 min of measurement. So for me, this will not exclude the possibility of pollens arriving in a batch at JFJ, therefore explaining this higher concentration. Also, pollen concentration measured at Bern may not be the representative concentration of pollens arriving at JFJ as another site (Visp) at a roughly similar distance from JFJ reported half of the concentration around the same day (April 20th ?).**

Based on the back trajectory analysis we do not believe that the pollen concentration in Bern could have contributed to the INP concentration peak at JFJ during April 20-25 because no trajectories passed over Bern (see Figure 1 and 2 below). In addition, some air masses could come from the south of JFJ and as far south as north of Italy, or other regions north of JFJ within Switzerland as far as south of Germany (see Fig. 2) which we cannot exclude. We have modified the manuscript on lines 264-277 in the revised manuscript to clearly state this possibility.

**C) Air mass origin. I wonder if an analysis of the air mass origin could help in understanding this spring peak of INP. Where the air mass is coming from during the INP peak? Could this air mass have collected pollen from somewhere in Europe? Could pollen be transported from further away than Bern or Visp (like SDE)? The authors state that an anti-cyclone was present until April 16th. Could it have influenced the non-transportation of pollen to JFJ at that time (low INP)? Could the peak of INP arriving just after the end of the anti-cyclone be a result of transportation of air mass from mainland Europe (which was full of pollen)?**

Based on the reviewer comment, we conducted a back trajectory analysis for April 20th (Figure 2) and 22nd (Figure 1). We modified the manuscript to state that pollen fragments could come from the south of JFJ and north of Italy or north of Switzerland in line with the air mass trajectories shown (see line 264 revised manuscript).

Furthermore, since there was a second peak in pollen concentrations in Bern around April 19/20, which means other parts of Switzerland and Europe likely had increased pollen concentrations, we include this as a possibility for pollen fragment transport to the JFJ in the revised manuscript (see lines 266-278 and 274-275). We refrain from making much stronger statements than currently stated in the revised manuscript for two reasons: first, there are no direct pollen measurements are JFJ as we stated in line 257 – 258 initial manuscript, line 266 revised manuscript. Second, we concluded using the same data set that dust contributed to ~98% of the INP concentrations at JFJ for all the data measured as reported in [*Brunner et al.*, 2021].

[Figure]

*Figure 1. Back trajectories (HYSPLIT, NOAA)[Rolph et al., 2017; Stein et al., 2015] ending at JFJ at 12.00 UTC on April 22nd showing no air masses arrive from Bern and Air masses arriving from Visp would only have passed over Visp after April 19, when pollen concentrations in Visp had already declined. The colours represent different trajectories starting every 3 hours going backwards. (https://www.ready.noaa.gov/HYSPLIT_traj.php)*

[Figure]

*Figure 2. Back trajectories (HYSPLIT, NOAA) [Rolph et al., 2017; Stein et al., 2015]arriving JFJ on April 20th at 12.00 UTC showing no air masses arriving from Bern or Visp, suggesting that these two locations would not contribute pollen fragments to the INP concentrations peak observed at JFJ during April 20th to 25th. The colours represent different trajectories starting every 3 hours going backwards. (https://www.ready.noaa.gov/HYSPLIT_traj.php)*

**Small comments:**

**It would be good to have Brunner et al. accepted 2021 in ACP to use the same notation as in this paper (if it is still possible to edit the manuscript). For example table 1 in both paper show the same results but with different notation.**

We thank the reviewer for the comment and agree that it would be good to have the same notation in this work and in *Brunner et al.* [2021]. And we agree that the notation from this work should be used, however, we cannot change the notation in the other manuscript anymore since it has already been published last year.

The alternative would be to change the notation in this work, but we explain why we refrain from doing so.  In Brunner et al. we only had "SDE in FT" or "SDE with BLI", whereas here we would have "BG in FT" and "BG with BLI" in addition. In our opinion, using the same notation as in *Brunner et al.* [2021] reduces the legibility. One side remark: The tables do not show identical data. This work shows data from Feb 7, 2020 to Jan 31, 2021 whereas *Brunner et al.* [2021] shows data from Feb 7 – Dec. 31. Because there were no SDE in Jan 2021, the statistics for SDE periods are identical.

**P3L87-88 "(CPC), TSI 3772, lower cut-off size: 14 nm) and size distribution (scanning mobility particle sizer (SMPS); optical particle sizer (OPS)" What is the size range of the SMPS (which is then used to calculate N90) and the size range of the OPS?**

We have added the respective size ranges of the SMPS and the OPS to the revised manuscript (see lines 88-89 in revised manuscript)

**Fig 3: "with the Q10% of PFT of a given day" what is the right axis BLI/FT %? I m a bit confused about how to read this scale. I am assuming that data close to BLI correspond to 0% and close to FT correspond to 100%. is that correct?**

The data close to the BLI label correspond to 0% (or a small percent of FT), i.e. the data close to the bottom (BLI) is predominantly when JFJ was in the boundary layer, and close to the top is when JFJ was predominantly in the free troposphere. We add the respective percentages to the axis in the revised manuscript.

**Fig 3: What is the meaning of the dash black line around mid-April in panel a) and b).**

The dashed black line indicates the peak in INP concentration for April to show that it does not align with the peak observed in Bern or Visp. We have added the definition of the dashed black line to the figure caption in the revised manuscript.

**Reference Schneider, J. et al. 2020 is from ACPD, Schneider et al. 2021 is the final version. Please correct in the text and in the reference list.**

Corrected

**Reference Brunner et al. 2021 in ACPD might be available at the time of the publication of this article.**

Corrected

**if other references of manuscripts in "preparation" are now available, please add them.**
**Powered by**

We have updated Wieder et al., in prep to the final version *Wieder et al.* [2022]

**References**

Brunner, C., B. T. Brem, M. Collaud Coen, F. Conen, M. Hervo, S. Henne, M. Steinbacher, M. Gysel-Beer, and Z. A. Kanji (2021), The contribution of Saharan dust to the ice-nucleating particle concentrations at the High Altitude Station Jungfraujoch (3580 m a.s.l.), Switzerland, *Atmos. Chem. Phys.*, *21*(23), 18029-18053, doi:10.5194/acp-21-18029-2021.

Rolph, G., A. Stein, and B. Stunder (2017), Real-time Environmental Applications and Display sYstem: READY, *Environmental Modelling & Software*, *95*, 210-228, doi:https://doi.org/10.1016/j.envsoft.2017.06.025.

Stein, A. F., R. R. Draxler, G. D. Rolph, B. J. B. Stunder, M. D. Cohen, and F. Ngan (2015), NOAA's HYSPLIT Atmospheric Transport and Dispersion Modeling System, *Bull. Amer. Meteorol. Soc.*, *96*(12), 2059-2077, doi:10.1175/BAMS-D-14-00110.1.

Wieder, J., C. Mignani, M. Schär, L. Roth, M. Sprenger, J. Henneberger, U. Lohmann, C. Brunner, and Z. A. Kanji (2022), Unveiling atmospheric transport and mixing mechanisms of ice-nucleating particles over the Alps, *Atmos. Chem. Phys.*, *22*(5), 3111-3130, doi:10.5194/acp-22-3111-2022.